# Omics Approaches in Adipose Tissue and Skeletal Muscle Addressing the Role of Extracellular Matrix in Obesity and Metabolic Dysfunction

**DOI:** 10.3390/ijms22052756

**Published:** 2021-03-09

**Authors:** Augusto Anguita-Ruiz, Mireia Bustos-Aibar, Julio Plaza-Díaz, Andrea Mendez-Gutierrez, Jesús Alcalá-Fdez, Concepción María Aguilera, Francisco Javier Ruiz-Ojeda

**Affiliations:** 1Department of Biochemistry and Molecular Biology II, School of Pharmacy, University of Granada, 18071 Granada, Spain; augustoanguitaruiz@gmail.com (A.A.-R.); mireia251019@gmail.com (M.B.-A.); jrplaza@ugr.es (J.P.-D.); andmengut@gmail.com (A.M.-G.); fruizojeda@ugr.es (F.J.R.-O.); 2Instituto de Investigación Biosanitaria IBS.GRANADA, Complejo Hospitalario Universitario de Granada, 18014 Granada, Spain; 3Institute of Nutrition and Food Technology “José Mataix”, Center of Biomedical Research, University of Granada, Avda. del Conocimiento s/n., 18016 Granada, Spain; 4CIBEROBN (CIBER Physiopathology of Obesity and Nutrition), Instituto de Salud Carlos III, 28029 Madrid, Spain; 5Children’s Hospital of Eastern Ontario Research Institute, Ottawa, ON K1H 8L1, Canada; 6Department of Computer Science and Artificial Intelligence, University of Granada, 18071 Granada, Spain; jalcala@decsai.ugr.es; 7RG Adipocytes and Metabolism, Institute for Diabetes and Obesity, Helmholtz Diabetes Center at Helmholtz Center Munich, Neuherberg, 85764 Munich, Germany

**Keywords:** obesity, adipose tissue, extracellular matrix, skeletal muscle, genetics, epigenetic, transcriptomic

## Abstract

Extracellular matrix (ECM) remodeling plays important roles in both white adipose tissue (WAT) and the skeletal muscle (SM) metabolism. Excessive adipocyte hypertrophy causes fibrosis, inflammation, and metabolic dysfunction in adipose tissue, as well as impaired adipogenesis. Similarly, disturbed ECM remodeling in SM has metabolic consequences such as decreased insulin sensitivity. Most of described ECM molecular alterations have been associated with DNA sequence variation, alterations in gene expression patterns, and epigenetic modifications. Among others, the most important epigenetic mechanism by which cells are able to modulate their gene expression is DNA methylation. Epigenome-Wide Association Studies (EWAS) have become a powerful approach to identify DNA methylation variation associated with biological traits in humans. Likewise, Genome-Wide Association Studies (GWAS) and gene expression microarrays have allowed the study of whole-genome genetics and transcriptomics patterns in obesity and metabolic diseases. The aim of this review is to explore the molecular basis of ECM in WAT and SM remodeling in obesity and the consequences of metabolic complications. For that purpose, we reviewed scientific literature including all omics approaches reporting genetic, epigenetic, and transcriptomic (GWAS, EWAS, and RNA-seq or cDNA arrays) ECM-related alterations in WAT and SM as associated with metabolic dysfunction and obesity.

## 1. Introduction

Healthy adipose tissue expansion in obesity depends on the extracellular matrix (ECM) remodeling and reorganization to provide enough space for the enlargement of adipocytes (hypertrophy) and for the generation of new ones through adipogenesis from the precursor cells (hyperplasia). This process involves the formation of new blood vessels, which is crucial to maintain healthy adipose tissue expandability because the failure of this results in necrosis adipocytes, and hypoxia, which triggers chronic, low-grade inflammation, fibrosis, and lastly insulin resistance [1,2,3,4]. Regardless of the manner of expansion, variations in ECM composition substantially affect the biomechanical properties of a tissue, as is clearly detected in fibrotic diseases [5]. Fibrosis is characterized by an increase in fibril-forming ECM proteins, and one of the major adhesion receptors implicated in the regulation of this process are integrins, and its signaling in the diet-induced obese state are associated with insulin resistance in adipose tissue [4,6,7,8]. The inability to develop a healthy adipose tissue expansion under the excess of calories drives an ectopic fat deposition in visceral depots, liver, SM, and other cell types [9]. Furthermore, this process produces adipose tissue dysfunction that leads to changes in the ECM. Beyond these effects in adipose tissue, fat deposition in SM incites similar effects to the progressive muscle weakening associated with aging, such as insulin resistance and metabolic abnormalities [10]. Thus, besides the effect of muscle fat accumulation on insulin sensitivity, muscle lipid accumulation also promotes changes in muscle ECM [11]. Genes such as collagens, fibronectin, proteoglycans, and connective tissue growth factor are upregulated under acute lipid oversupply into the SM of healthy individuals [12]. Moreover, collagen I and collagen III are higher in SM from obese and T2D individuals [13].

In the past decade, genome-wide association studies (GWAS) have successfully identified thousands of genetic variants underlying susceptibility to complex diseases given its ability to rapidly scanning markers across the whole genome. However, the results from these studies often do not provide evidence on how the variants affect downstream pathways and lead to the disease. Hence, in the post-GWAS era, the greatest challenge lies in combining GWAS findings and functionally characterize the associations. Epigenetics is the study of heritable phenotype changes that do not involve alterations in the DNA sequence and its mechanisms control gene activity and the development of an organism [14,15]. In particular, DNA methylation is an epigenetic mechanism that modifies the function of genes and affects gene expression, and aberrant DNA methylation has been found to be associated with various complex human diseases, including obesity. EWAS have been useful in identifying disease-associated epigenetic marks for screening high-risk populations, and EWAS have identified several differentially methylated CpG regions related to body mass index (BMI) and other parameters of obesity. Henceforth, DNA methylation provide evidence on how molecular dysregulation can affect variety of gene types and omics approach addressing epigenetic mechanism has been increased in the last years. Indeed, DNA methylation plays an important role in adipogenesis, and adipose tissue expansion [16], and changes in DNA methylation patterns of SM in specific genes may affect to whole-body insulin sensitivity in obesity [17]. Therefore, the aim of the present review is to include the omics approach addressing genetic (GWAS), epigenetic (EWAS), or gene expression (RNA-seq and cDNA arrays) mechanisms in WAT and SM in order to understand the molecular mechanism of ECM in metabolic dysfunction associated with the obesity development.

## 2. Methodology

### 2.1. Systematic Review Strategy

This systematic review was performed according to the guidelines described by the Preferred Reporting Items for Systematic Review and Meta-Analysis (PRISMA) statement [18,19]; definition of the research question, literature search, data collection, evaluation, comparison, and synthesis, as well as critical analysis and findings presentation, showing the strengths and weakness of the studies analyzed. A bibliographic search strategy was conducted to identify all studies reporting ECM genetics/epigenomics/transcriptomics findings for obesity and its associated cardiometabolic phenotypes in humans. For that purpose, we considered both in vitro and in vivo studies and restricted our search to three obesity-related tissue types (blood, adipose, and SM). The electronic database consulted was NCBI/PubMed, which was automatically and iteratively mined using R environment (https://www.r-project.org/, accessed on 15 December 2020). Particularly, we employed an R package called “*easyPubMed*” (https://CRAN.R-project.org/package=easyPubMed, accessed on 15 December 2020), which allows: (1) Query NCBI Entrez IDs and retrieve PubMed records in XML or text format, (2) Process PubMed records by extracting and aggregating data from selected fields. Prior to the search process, we identified all genes annotated in the GO and KEGG databases with terms related to the ECM (for more details regarding the ECM terms considered, please see Appendix A). As a result, we identified 2186 different items. Then, we employed three Boolean searches (Appendix A), corresponding to each omics of interest (genomics, epigenomics, and transcriptomics), which were iteratively run 2186 times, one time for each of the 2186 *loci*. Thanks to this strategy, we were able to map all research studies reporting omics findings for any of the genes that are currently known to be involved or related to the ECM. Findings were therefore identified at the locus level and organized according to ECM gene ontology terms (focusing on biological process GO and KEGG terms exclusively). By organizing associations according to the ECM structures or processes affected, we further aimed to generate new insights into the molecular basis of obesity and metabolic alterations. 

### 2.2. Study Selection

In total, 634 studies were collected through the identification of records in PubMed following the different search strategies. Moreover, 31 additional articles, referenced in any chosen record and not presented in Boolean searches, were forced to inclusion, since they accomplish inclusion criteria. Inclusion criteria were studies written in English; published from April 1999 to October 2020; the presence of the selected search words in the title, abstract, or as keywords, depending on the algorithm; obesity or cardiometabolic alterations as the main outcome; investigation performed in humans (studies in animals and in vitro models were also considered, whether their aims were referred to the investigation of the human gene considered); whole-genome array-type approach; report information for the locus included in the search equation. Exclusion criteria were review articles; duplicates; and studies focused on a different gene or technology with regard to those selected for the search strategy. 

The selection of studies began by screening titles, abstract and/or main manuscript for inclusion, generating a reference list of intriguing articles. A total of 345 studies were assessed and selected following the eligibility evaluation, the implementation of the exclusion criteria and fulfill inclusion criteria, including in Section 4, Section 5, and Section 6 of the present manuscript. A summary of the stages can be observed as a PRISMA flow diagram in Appendix A.

## 3. Current Evidence of Adipose and SM ECM Remodeling in Obesity and Metabolic Dysfunction

Adipose tissue ECM is composed mainly of different types of collagens (I, II, III, and IV), fibronectin, and a small amount of laminin [20,21]. However, several components, such as A disintegrin and metalloproteinase domain-containing protein (ADAMs), osteopontin (OPN), hyaluronan (HA), thrombospondins (THBS1), matrix metalloproteases (MMPs), and tissue inhibitor of metalloproteinases (TIMPs), play an important role in the ECM remodeling and adipose tissue function [22]. Integrins play an important role in ECM remodeling in adipose tissue, and they are the main class of tissue receptors implicated in cell-extracellular matrix interactions and cell adhesion. A recent study reveals that integrins, as the main cell surface involved in cell adhesion, interact with insulin receptor in adipocytes modulating WAT insulin sensitivity, demonstrating a new cell-matrix interaction in adipocytes [8]. Therefore, it seems that integrins and insulin signaling interact with each other and play an important role in ECM remodeling in adipose tissue. The cell surface transmembrane glycoprotein CD44 is ubiquitously expressed and it binds to the ECM, mainly OPN and HA. 

Collagens are a major abundant fibrous protein in the ECM and adipocytes primarily produce collagen, though the preadipocytes, endothelial cells, and stem cells are also able to produce it. Mature adipocytes stock energy as triacylglycerol’s, and this process provokes robust mechanical stress, which is relocated from the outside to the inside of the cell spending a lot of energy on the maintenance of the ECM [6,23,24]. Adipose tissue from obese mice (C57BL/KsJ-lepr*^db^*/lepr*^db^*) undergoing a high-fat diet (HFD) (40% fat) exhibit a higher collagens content such as collagen I, III, and VI, and increased expression of procollagens I, III, V, VI, and VIII [25]. 

MMPs family members might be characterized into soluble collagenases (MMP1, -8, and -13), gelatinases A and B (MMP2 and -9), stromelysin-1, 2 and 3 (MMP3, -10, and -11), matrilysin-1 and -2 (MMP7 and -26), membrane-type MMPs (MT-MMPs) (MMP14, -15, -16, -17, -24, and -25), and elastase (MMP12) [26]. Endothelial cells, pericytes and podocytes, fibroblasts, and myofibroblasts, and macrophages secrete MMP-2 and -9 [27]. MMP-3, MMP-11, MMP-12, MMP-13, and MMP-14 levels are upregulated in abdominal WAT, while MMP-7, MMP-9, MMP-16, MMP-24, and TIMP-4 were downregulated in mice (75% C57/Bl6:25% 129Svj genetic background) under HFD feeding (42% Kcal as fat) [28]. On the other hand, MMP-2 and MMP-9 activity are decreased in WAT from the insulin-resistant Wistar rats fed upon 30% sucrose-rich diet, and no modifications were related to MMP plasma activity [29]. TIMPs can perform as endogenous inhibitors of MMPs that are reliable for destroying additional ECM, it is uncertain whether the useful effects of augmented TIMP or ADAMTS activities are solely due to the inhibited activity of MMPs and augmented ECM stability [30]. In particular, plasma concentrations of TIMP1 and TIMP2 are higher in patients with metabolic syndrome and diabetes [31], and gene deletion of TIMP-2 in C57Bl/6 mice induces obesity in HFD feeding (60% fat) [32]. However, exercise is rather supportive in TIMP-2 modulation, improving insulin sensitivity [33].

SM is one of the most dynamic and plastic tissues of the human body and contains around 40% of total body weight and comprises 50–75% of all body proteins [34]. Like WAT, SM ECM remodeling is required for a proper homeostasis, which implicates a constant regulation of the synthesis vs. degradation of the main components. ECM plays an important role in muscle fiber force transmission, maintenance, and repair. The muscle ECM distribution is unique and consists of several components including endomysial (around the muscle cell), perimysial (around groups of muscle cells), and epimysial (around the whole muscle) connective tissues [35,36]. Collagen is the main protein in SM ECM, which represents between 1–10% of muscle mass dry weight [35]. There are several types of collagen-containing types I, III, IV, V, VI, XI, XII, XIV, XV, and XVIII expressed during the muscle development, although the types I and III are the predominant ones in adult endo-, peri-, and epimysium [37]. Muscle basement membrane consists primarily of a type IV collagen network, but types VI, XV, and XVIII are also present [38]. Proteoglycans are another muscle ECM component that belong to the family of small leucine-rich proteoglycans containing a core protein with attached GAG chains and include decorin, biglycan, fibromodulin, and lumican. A recent study has identified by ECM-specific mass spectrometry-based proteomics technique distinct signatures of HFD-induced protein changes between SM and liver in C57BL/6J mice. In particular, SM collagens isoforms increased upon HFD feeding (60% fat) and collagen 24α1 is associated with insulin resistance in SM. Moreover, they found that collagen 24α1 is highly expressed in visceral adipose tissue (VAT), but not in SAT, of obese diabetic subjects, indicating a potential pathogenic role of that collagen in obesity and T2D [39].

Alike adipose tissue, in normal SM, a delicate balance occurs between enzymes responsible for ECM synthesis and their inhibitors. Thus, the turnover of ECM is required for cell migration, myotube formation, and reorganization of the matrix during muscle adaptation. Briefly, MMPs are the major determinants in ECM remodeling and participate in the degradation of ECM components [40]. MMP-2 and MMP9 are the subclasses of metalloproteases involved in the development of obesity and insulin resistance [36,41], and indeed MMP-9 plasma levels are higher, despite lower protein levels in SM under HFD (60% fat) fed C57BL/6J mice [41]. Furthermore, MMP-9 is released in response to myostatin, which promotes a turnover of ECM components in SM. Mostly, the balance between MMP and TIMPs is certainly critical to maintain a repair process in the context of ECM. Besides, MMP14 content is significant in SM due to the function in the activation of the MMP2 [36]. Therefore, MMPs and TIMPs are relevant to maintain the ECM homeostasis in muscle. 

Several genes implicated in ECM remodeling in adipose tissue are also involved in SM. Integrins, which play an important function in adipose tissue [8], are implicated in myogenesis [42] and adipogenesis [43]. Thus, integrin striated muscle-specific integrin β1-deficient C57Bl6/J mice exhibit decreased in AKT Ser-473 phosphorylation, glucose uptake, and glycogen synthesis, supporting the role between integrin and insulin signaling in SM [44,45]. Other ECM components such as proteoglycans [46,47], fibronectin [48,49], hepatocyte growth factor (HGF) [50,51], epidermal growth factor (EGF) [52,53], insulin-like growth factor(IGF)-1 [54,55], IGF-2 [56,57], and MMPs [58,59], among others, play crucial roles in myogenesis and adipogenesis. Hence, understanding the inter-tissue connection between WAT and SM in some common genes implicated in ECM remodeling could help to develop strategies against the metabolic disturbances in obesity. The specific molecular mechanism of ECM remodeling has been reviewed elsewhere [4,60,61].

Overall, obesity drives changes in adipose and muscle ECM function and remodeling that may cause important metabolic alterations in the local and systemic metabolism. In addition, genetic variations and epigenetic changes, and in the end, the gene expression patterns can affect to those processed, which will be described in the next sections. 

## 4. Whole-Genome Genetic Approaches Reveal the Implication of ECM in Obesity and Metabolic Dysfunction 

In the present section, we gathered all literature findings from GWAS experiments in which sequence variation for ECM-related *loci* has been linked to obesity or metabolic dysfunction, although associations do not mean causative process. Findings were identified at the *locus* level and organized according to ECM gene ontology terms (focusing on biological process terms exclusively) (Table 1). By organizing GWAS associations according to the ECM structures or processes affected, we aimed to generate new insights into the molecular basis of obesity and metabolic alterations. In the next sections, these insights will be further integrated along with additional molecular data layers (such as associations derived from epigenomics and transcriptomics) in order to be functionally characterized.

From this search, 80 entries at the locus level were collected, corresponding to 67 unique scientific articles. After removing all entries not asserting our inclusion criteria, 61 entries finally remained (corresponding to 52 scientific articles). These entries mapped 15 gene ontology terms that were merged into broader biological categories according to expert knowledge: (I) Collagen-containing ECM, (II) Cell adhesion, (III) Endothelial cell migration, and (IV) ECM assembly and disassembly. These 4 biological categories refer to structures and processes of great importance in ECM with a direct connection to adipose tissue dysfunction, and lastly to insulin resistance. Bellow, results in Table 1 are summarized according to each of these categories.

The *collagen-containing extracellular matrix* category comprise a range of proteins (especially collagens and glycosaminoglycans, mostly as proteoglycans) that provide not only essential physical scaffolding for the cellular constituents but can also initiate crucial biochemical and biomechanical cues required for tissue morphogenesis, differentiation and homeostasis. These proteins are the main ECM component in adipose tissue and their excessive accumulation in obesity is the main cause of tissue fibrosis, rigidity, and finally insulin resistance. A total of 12 GWAS-reported *loci* were grouped into this category after the search (*ADIPOQ*, *APOE*, *CBLN4*, *COL4A1*, *COL6A5*, *CTSS*, *F12*, *F13A1*, *FRAS1*, *ITIH4*, *SGCZ,* and *SSPN*) [64,65,66,69,70,71,73,78,79,80,84,85,86,87,89,92,93,94,95,96]. From these, it highlights structural collagen components such as *COL4A1*, *COL6A5* and other types of structural ECM components such as *FRAS1*, *SGCZ,* and *SSPN*, for which SNPs and CNVs have been directly associated with higher risk of obesity or increased anthropometry measurements in diverse ethnic populations (at genome-wide significance levels P < 5 × 10^e−8^) [62,67,68,72,74,75]. Furthermore, it is also remarkable the association of the gene *cathepsin S* (*CTSS*), known to degrade several components of the ECM, which is produced by human adipocytes and increased in obesity. A novel SNP within this gene, the rs2230061, was associated with Fat Body Mass after the adjustment of lean body mass (P = 3.57 × 10^e−8^) at the genome-wide significance level.

The *cell adhesion* category comprised any process by which cells form contacts with each other or with ECM components through different receptors. Particularly, this category included *loci* mapping GO terms such as *adherens junctions*, *cell adhesions involved in heart morphogenesis*, *cell adhesion molecule binding*, *cell-matrix adhesions*, *focal adhesions,* and *regulation of cell adhesions*. Among others, this section included GWAS-reported associations mapping proteins such as *Cadherins*; a group of transmembrane glycoproteins that mediate intercellular adhesion in the presence of extracellular calcium and play important roles in cell–cell adhesions. Likewise, there were also matrix metalloproteinases (such as *ADAMs*), with disintegrin-binding regions, that can interact with integrins and mediate cell–ECM interactions. As it can be seen in Table 1, a total of 21 GWAS-reported *loci* were gathered in the category (*ADAM23, ADAMTS9, BCL6, C2CD4A, CD36, CDH12, CDH18, CELSR2, COBLL1, CSK, FLRT2, IDH1, IGF1, IGF1R, INSR, LEP, NRXN3, OLFM4, PFKP, RREB1,* and *TCF7L2*). Among them, we differentiate between *loci* exclusively associated with obesity and others, which also presents associations with insulin resistance or T2D (Table 1). For the *loci* annotated in the GO term *adherens junctions*, it outstands the *locus TCF7L2*, which is a transcription factor acting downstream of *B-catenin* and involved in blood glucose homeostasis. Interestingly, *TCF7L2* is one of the top *loci* associated with increased T2D risk in a wide range of studies [90]. Within the *adherens junctions* GO term, it is also interesting the association reported for the insulin receptor gene *INSR*, for which SNPs have been identified as functional variants affecting gene expression directly or indirectly via epigenetic (CpG-methylation) alterations [90]. Among the 21 GWAS-reported *loci* that were annotated into this category, we also reported two genes strongly related to each other (*IGF1, IGF1R*). SNPs within these genes have been associated both with weight-loss variability (in response to bariatric surgery intervention) and with higher indexes of IR in non-diabetic subjects [88]. This is interesting, since both *loci* mediate the production of adhesion molecules by endothelial cells and monocyte adhesion onto the vascular endothelium in response to the hyperinsulinemic state. Therefore, they probably exert a strong contribution to the pathogenesis of atherosclerotic disease in diabetes and obesity. In relation to matrix metalloproteinases, it outstands the *ADAM23, ADAMTS9*, for which SNPs have been associated with increased BMI and metabolic alterations (mainly lipid and glucose traits), respectively [75,76,77]. Finally, we also reported GWAS-associations within cadherin-type proteins such as (*CDH12, CDH18,* and *CELSR2*), associated with obesity and CRP serum levels, respectively [81,82,83]. 

In the *Endothelial Cell migration* category, we gathered all GWAS *loci* regulating the orderly movement of endothelial cells into the ECM to form an endothelium. Particularly, increased angiogenesis is a consequence of insulin resistance in adipose tissue. As a result, this category included a total of 6 GWAS-reported *loci* (*MAP2K3, MAP2K5, MET, PPARG, PROX1,* and *SIRT1*). Among them, it highlights (*MAP2K5* and *PPARG*) with a negative effect on the endothelial cell migration, for which SNPs have been repeatedly associated with obesity in several populations (especially in Asians) [63,91,99,100,101,102,103,104,106,107]. The rest of these *loci* were involved in a positive regulation of the endothelial cell migration, and their SNPs have been associated not only with an increased risk of obesity but also with T2D or gestational diabetes mellitus [97,98,105,108,109,110].

Finally, in the *ECM assembly and disassembly* category, we grouped all GWAS-reported *loci* involved in any process that results in the assembly, arrangement of constituent parts, or disassembly of the ECM. These processes are of special importance in obesity and metabolic dysfunction given the described mechanisms associated with disturbed ECM remodeling in adipose tissue expansion. A total of 5 GWAS *loci* (*GP2*, *IL6*, *LINGO2*, *SH3PXD2B,* and *SLC2A10*) were gathered into this category. From these, all showed strong associations with obesity (at genome-wide significance level or validated in independent samples) [104,111,112,114], except the *SH3PXD2B*, for which no evidence was found in two GWA samples [113].

## 5. Whole-Genome DNA Methylation Approaches Reveal the Implication of ECM in Obesity and Metabolic Dysfunction

In this section, we assembled all literature findings from EWAS experiments revealing the implication of ECM in obesity and metabolic dysfunction. We collected and organized according to ECM gene ontology terms, specifically focusing on biological process terms, but associations do not mean causative process (Table 2). In this search we initially collected 47 scientific articles, and after removing all entries not asserting our inclusion criteria, 23 finally remained. Different cell signaling pathway categories sorted these articles: *collagens*; *cell adhesion*; *endothelial cell migration*; *regulation of cytoskeleton*; and *ECM assembly and disassembly*. These categories refer to processes and cell-matrix interactions in ECM, which have been associated with metabolic dysfunction in the adipocytes, and lastly in the whole adipose tissue and SM. Bellow, results from Table 2 are summarized according to each of these categories. 

The *collagen-containing ECM* category included 6 EWAS-reported genes derived from our search (*ADIPOQ, COL11A2, COL23A1, LGALS3BP, PLG,* and *S100A4*). Although collagens are the main ECM proteins in adipose tissue, the methylation of some other genes in adipose tissue might also present some impact in fibrosis, stiffness, and adipocyte dysfunction, and ultimately in insulin resistance. In the case of the adipokine *ADIPOQ* (Adiponectin, C1Q, and Collagen domain containing), which it is secreted by adipocytes regulating insulin sensitivity, Perfilyev et al. (2017) showed that its mean methylation was increased in SAT after saturated fatty acid (SFA) overfeeding, and that it may contribute to the decreased adiponectin concentrations commonly observed in obese individuals [115]. Fibrosis, tissue inflammation, and insulin resistance are linked in adipose tissue, however, the causality direction of such relationship seems to be unclear [138]. On this matter, a recent study has reported that insulin resistance promotes accumulation of M1 macrophages and fosters inflammation [139]. In obese WAT, both hypoxia and inflammation induce a pathological expansion of ECM with macrophages recruitment and increased collagens deposition. Interestingly, epigenetic mechanisms could participate in this regulation of adipose tissue morphology associated with ECM remodeling. In the case of *COL11A2*, two different studies have evaluated the contribution of methylation levels in VAT either in obese individuals with metabolic syndrome or in individuals with insulin resistance. From these, Guenard et al. (2017) identified a higher degree of methylation for 2 *COL11A2* CpGs in the VAT of obese men with metabolic syndrome. Interestingly, these 2 CpGs were reported to interact with SNPs and affect fasting plasma glucose levels, providing a potential biological mechanistic insight on the development of metabolic syndrome [116]. Likewise, *COL11A2* exhibited higher methylation levels in the VAT of individuals with insulin resistance compared to normal insulin sensitivity controls [117]. Similarly, *COL23A1* showed lower methylation levels in morbidly obese patients, and these levels were associated with the activity of the enzyme nicotinamide N-methyltransferase, which is a major methyltransferase associated with BMI and insulin resistance. Those results were confirmed in two different cohorts and validated in a weight loss intervention study, linking low methylation levels of *COL23A1* with adipose tissue dysfunction [118]. Within this collagens category, it also outstands the lectin galactoside-binding soluble 3 binding protein (*LGALS3BP*), a macrophage inflammatory marker, for which a higher methylation in SAT was associated with adiposity [119]. On the other hand, Roon et al. (2015) found that plasminogen (*PLG*) was differentially methylated in human SAT from subjects with T2D compared to non-diabetic controls, and its methylation levels were negatively correlated with age and expression [120]. Within this category, it is also important to highlight the *S100A4* metastasis-associated protein, for which our research group have reported an association with insulin resistance and WAT dysfunction in prepubertal populations. Interestingly, we reported how the change in plasma *S100A4* levels accompanies longitudinal trajectories of insulin resistance in children and how the methylation levels for two-enhancer-related CpG sites of the *S100A4* region (cg07245635 and cg10447638) perfectly correlate with insulin resistance biomarkers at the prepubertal stage [121]. 

Secondly, within the *cell adhesion* category, we included GO terms such as *adherens junctions*, *cell adhesion molecule binding*, *cell matrix adhesion*, *regulation of cell adhesion*, and *tight junctions*. As can be seen in Table 2, a total of 15 EWAS-reported genes were gathered in this category (*ADAM2*, *BAG3*, *CCL2*, *CD36*, *CD44*, *CDK5*, *CLDN1*, *COL5A1*, *HLA-DPB1*, *IGFBP2*, *LEP*, *MAPK7*, *SFRP1*, *SPP1*, and *TCF7L2*). Annotated with the GO term *adherens junction*, *TCF7L2* is one of the most important *loci* in the category. Interestingly, *TCF7L2* is also a GWAS-reported locus associated with T2D and glucose homeostasis. Here, we found that differential DNA methylation in *TCF7L2* is also associated with obesity and TD2 according to three different studies. Particularly, it has been described that the methylation of *TCF7L2* is lower in SAT from monozygotic twin pair’s unrelated subjects with T2D compared to control subjects [122]. Similarly, lower DNA methylation of *TCF7L2* in SAT was associated with BMI [123]; and associated with insulin resistance in adipose tissue samples from lean and obese patients pre- and post-Roux-en-Y gastric bypass. In this study, *TCF7L2* genetic variation was highlighted as a cause for the changes in methylation and with a direct implication in the development of T2D [124]. Mapping the Go term *cell adhesion molecule binding*, we find the *SPP1*, which encodes the inflammatory cytokine osteopontin. This *locus* was found to be more methylated in human SAT from subjects with T2D in comparison to non-diabetic controls, and this methylation has previously been linked to a higher expression level in adipose tissue and inflammation [120]. In concordance with these results, Nilsson et al. (2014) showed a different DNA methylation pattern of *SPP1* between the SAT of monozygotic twin pairs unrelated subjects with T2D and control subjects [122], supporting the biological validation of described genetic-epigenetic modifications of *SPP1* in adipose tissue. Another study showed that *BAG3*, an anti-apoptotic protein and an indicator of cellular stress, is differentially methylated after short- and long-term weight loss programs in SAT of healthy participants, resulting in a downregulation of gene expression and demonstrating a response during weight loss; however, no changes were found in VAT [125]. Another study showed higher DNA methylation of *COL5A1* in the VAT of women with insulin resistance, revealing an epigenetic regulation of *COL5A1* and its implication in pathways related to integrin cell interactions and insulin signaling in VAT [126]. In this sense, *ADAM2*, a type of disintegrin regulating cell-matrix interactions, exhibited lower methylation in VAT of insulin resistance subjects than in healthy individuals, revealing a potential epigenetic regulation of adipose tissue dysfunction [117]. In the *cell matrix adhesion* GO term, we identified four genes (*CD36, SFRP1, CDK5, and CD44*). From them, the fatty acid translocase *CD36*, which is involved in lipid metabolism, was more methylated and less expressed in VAT among non-obese subjects and affects metabolism through several mechanisms [122]. Moreover, Andersen et al. (2019) reported that preadipocytes of VAT from obese and T2D subjects are transcriptionally different in response to differentiation in culture, compared to those of lean, showing impaired insulin signaling and a further transcriptomic shift towards altered adipocyte function. Thus, cultures with a lower expression magnitude of adipogenic genes throughout differentiation such as *CD36* were associated with DNA methylation at remodeling in genes controlling insulin sensitivity and adipocytokine signaling pathways, supporting the role of *CD36* in adipocyte function [128]. *SFRP1* is a modulator of Wnt/β-catenin signaling regulating adipogenesis, and its expression is reduced in obesity [140]. Epigenetic modifications of *SFRP1* were reported in mice overexpressing DNA methyltransferase enzyme in the adipose tissue, showing how a slight increase in the methylation of *SFRP1* associates with a marginal suppression of its gene expression and higher inflammatory markers, therefore linking epigenetic changes to inflammation in adipose tissue [129]. *CDK5*, which phosphorylate *PPARγ* in adipocytes, is highly methylated in SAT after five days of high-fat overfeeding in individuals who had low birthweight, indicating how short-term overfeeding influences methylation and gene transcription in adipose tissue [130]. The cell-surface glycoprotein *CD44*, involved in cell–cell interactions, exhibited higher methylation in VAT of insulin resistance than in healthy individuals, demonstrating that VAT from patients with disturbances in the insulin sensitivity presents a specific DNA methylation pattern [117]. In the GO term *regulation of cell adhesion*, the *loci HLA-DPB1*, *LEP*, *MAPK7*, *IGFBP2*, and *CCL2* were gathered. *HLA-DPB1*, which plays an essential role in obesity-induced adipose inflammation, is hypermethylated in human SAT and positively correlated with HbA1c, supporting the link between inflammation and systemic insulin resistance [120]. The *MAPK7*, which plays a pivotal role in proliferation and differentiation, is hypermethylated in SAT in response to SFAs compared to polyunsaturated fatty acids (PUFAs) intake, supporting that SFA overfeeding induces distinct epigenetic changes in human SAT and this might affect to ECM remodeling in the tissue [115]. *IGFBP2* is the second most circulating IGFBP secreted by liver and white adipocytes, and DNA methylation levels are higher in VAT, also SAT but lower compared to VAT, from subject with obesity, indicating a potential implication of *IGFBP2* in abdominal obesity [131]. We continuously show here that DNA methylation in ECM-associated genes influences inflammation in adipose tissue. Thus, Petrus et al. (2018) demonstrated that adipocytes of SAT from obese individuals’ exhibit a global DNA hypermethylation associated positively with gene expression of proinflammatory pathways. Besides, they observed that *SLC19A1* knockdown, a folate carrier, induced DNA hypermethylation of *CCL2* in the promoter-located cg12698626 in human adipocytes, further resulting in a higher inflammation in WAT and adipocyte dysfunction, finally contributing to the development of insulin resistance [132]. Within the GO term *tight junction*, claudin 1 (*CLDN1*), encoding an integral membrane protein and a component of tight junction strands, was strongly hypermethylated in human VAT and SAT of obese subjects compared to normal weight. In addition, *CLDN1* expression is lower in SAT compared to VAT from non-obese individuals, suggesting a functional role of *CLDN1* in VAT [127]. In relation to the *focal adhesion* GO term, one interesting study reported that adipocytes from overweight/obese people display decreased insulin sensitivity and reduced expression of *AKT2*, which it is associated with higher methylation at regulatory sites in the *AKT2* promoter. These findings indicate that healthy and non-diabetic subjects have already disturbed insulin signaling in the overweight state; it is worsened in healthy obese individuals and being only slightly aggravated in metabolically unhealthy obesity, consequently contributing to early defects in insulin action in adipocytes [133]. 

In the *endothelial cell migration* category, *PPARG* was hypermethylated in SAT [115,122], and VAT [127], and it was associated with obesity, indicating an important role of this gene in the reorganization of ECM. 

The *regulation of cytoskeleton* category included 3 EWAS-reported genes that were collected after the search (*ARHGEF1, ARHGEF4,* and *ARPC3*). Many of the cytoskeletal proteins found within the cell interact with extracellular matrix proteins, and linkage between cell surface receptors responsible for sensing the matrix and the cytoskeleton may be relevant to understanding the mechanism behind those responses. Actin-related protein 2/3 complex subunit 3 (*ARPC3*) is a gene recently linked to adipogenesis and lipid accumulation in obese people. A significant association was found between the CpG-SNP rs3759384 (C > T) and plasma triglyceride (TG) levels. In addition, the carriers of the rs3759384 T allele also showed a significant decrease in methylation levels of the *ARPC3* promoter-associated CpG site cg10738648 in VAT. Therefore, *ARPC3*, which is involved in cytoskeleton organization during adipose tissue expansion, might have an impact on the development of metabolic syndrome, acting through the regulation of DNA methylation in VAT [137]. Wang et al. (2018) described that the *rho guanine nucleotide exchange factor 1* (*ARHGEF1*) gene is hypermethylated in VAT from diabetic individuals compared with healthy controls. So far, those genes could be diabetes susceptibility genes epigenetically regulated in VAT [141]. 

Finally, in the *ECM assembly and disassembly* category, we collected all EWAS-reported *loci* involved in any process that results in the assembly, arrangement of constituent parts, or disassembly of the ECM. In this category, 5 EWAS-reported genes were collected (*PHOSPHO1, NOTCH1, CD4, IL6,* and *TNF*). Dayeh et al. (2016) identified that DNA methylation at the *PHOSPHO1* locus (cg02650017) in blood DNA is associated with a decreased risk for future T2D and positively correlated with HDL levels. Besides, they found a decreased DNA methylation of *PHOSPHO1* locus (cg02650017) in SM from diabetic versus non-diabetic monozygotic twins, suggesting a pivotal role of this gene in ECM in SM associated with metabolic disease [134]. Another study investigated the smoking-associated DNA methylation in adipose tissue, and they observed that smoking-associated differentially methylated regions in *NOTCH1* were significantly associated with measures of metabolic disease [135]. The *CD4* has been found to be differentially methylated in monocytes, and dendritic cells in SAT and negatively associated with android fat mass, confirming that higher leukocyte infiltration is directly associated with higher levels of adiposity. Consequently, epigenetic signatures could potentially be used as biomarkers for identifying leukocyte infiltration and risk of insulin resistance [136]. Likewise, Demerath et al. (2015) also reported higher DNA methylation in the *CD4* gene in SAT [119]. In the Perfilyev et al. (2017) study, they also found higher DNA methylation in SAT of *IL6* associated with the degree of weight increase in response PUFA overfeeding [115]. 

Overall, omics approaches have been robustly used to understand the implication of DNA methylation and its consequences at in the ECM remodeling in adipose tissue expansion, in relation to different cell signaling such as cell adhesion, collagens, and ECM assembly that can contribute to adipocyte dysfunction, and therefore, in the whole-body metabolism in obesity. 

## 6. Whole-Genome Gene Expression Approaches Reveal the Implication of ECM in Obesity and Metabolic Dysfunction 

Gene expression levels reflect the combined effect of a wide range of genomic modifications including point mutations, structural variants, and epigenetic changes. Furthermore, some specific mRNA is likely more closely reflecting the overall genomic effects than each type of variation separately [142]. In this context, gene expression patterns in adipose tissue and SM are used to study various aspects of systemic metabolism. Here, we will summarize the differential gene expression, by using microarrays or RNA-seq approaches, revealing the implication of the ECM in adipose tissue and SM in obesity and metabolic complications. Findings related to gene expression were also organized according to ECM gene ontology terms (Table 3). Terms are merged according to expert knowledge: *collagens*, *cell and focal adhesion*, *endothelial cell migration,* and *ECM assembly and disassembly*. 

In the *collagen-containing ECM* category are included 11 genes (*ASPN, COL24A1, COL6A1, COL6A2, EFEMP1, ELN, LOXL2, MMP8, SOD3, SPARC, and TGFB1)*. As abovementioned, collagens accumulation contributes to fibrosis in adipose tissue upon weight gain, however, the evolution of ECM remodeling in SAT after weight loss was not entirely understood. *ASNP* is highly expressed in adipocytes and it is associated with ECM, and higher baseline expression in SAT in individuals who successfully maintained weight loss was relevant for prevention of weight (re)gain. Interestingly, *ASPN* is a *TGFB1* inhibitor and the increased inhibition of *TGFB1* pathway by *ASPN* resulted in increased weight control. However, the ECM glycoprotein *SPARC* exhibited lower expression in individuals who had the greatest decreases in BMI. Those genes emphasize the potential role of tissue fibrosis in long-term weight control [143]. Another study has identified that maternal *COL24A1* variants, in particular, rs11161721 is the top one showing a significant genome-wide interaction with maternal pre-pregnancy overweight and obesity on preterm birth risk. Interestingly, in adipose tissue, the variant rs11161721 is significantly associated with altered *COL24A1* expression, providing new insights about a novel gene-maternal pre-pregnancy BMI interaction on preterm birth [144]. Some genes are also affected by rare CNVs in subjects with obesity, and thereby, this drives an alteration in gene expression of epidermal growth factor *EFEMP1,* which is involved in ECM remodeling in SAT, indicating that rare CNVs could be involved in the development of early-onset obesity [146]. Concerning bariatric surgery and weight control, a study conducted transcriptomic and histological characterization in SAT after the first year of bariatric surgery and, interestingly, they found higher degraded collagen accumulation in SAT after bariatric surgery along with fat mass loss and different gene expression of ECM components such as *COL3A1, COL6A1, COL6A2,* and *ELN.* Therefore, adipose tissue is able to adapt after a drastic weight loss, showing an increase in collagen degradation, but further studies are needed in order to follow up patients during long-term weight loss and elucidate the impact of adipose tissue remodeling in the systemic metabolism [145]. In relation to adipogenesis, by using an in vitro 3D culture system, Pellegrinelli et al. (2014) described in human adipocytes and decellularized material of adipose tissue (dMAT) from obese individuals a higher inflammatory and fibrotic gene expression of *LOXL2*, suggesting that fibrosis affects negatively to human adipocyte function via mechanosensitive molecules [147]. Li et al. (2010) identified that *Sod3*, concerned with oxidative stress, co-varies with the neuronatin (*Nnat*) promoter in mice adipocytes, which is an acute diet-responsive gene in adipose tissue affecting adipogenesis and metabolism [149]. 

Here, the cell adhesion category included 12 genes (ANXA1, COL5A1, EGFL6, FSTL3, ICAM1, IL1B, IRF1, JUN, LGALS12, PTPRJ, TCF7L2, and VCAM1). In this category, the GO terms adherents junctions, cell adhesion molecule binding, and regulation of cell adhesion are included (Table 3). Regarding inflammation and the term cell adhesion molecule binding, our research team showed that ANXA1, which is involved in the inflammatory process, is upregulated in the VAT of obese compared to normal weight prepubertal children [150]. In a different study from Arner et al. (2016), as it is described in the methylation section, 51 genes were reported as differentially expressed in VAT of insulin resistance obese women (e.g., COL5A1 and PTPRJ), revealing the different gene expression patterns in adipose tissue [126]. IL-1β, a leukocytic pyrogen activated by macrophages, is involved in the inflammatory process, and its expression in the regulation of ECM and cell adhesion has been examined in human adipocytes. The incubation with IL-1β in adipocytes upregulates ICAM1 and VCAM1, indicating an effect on the expression of ECM and cell adhesion genes in human adipocytes, consistent with the derangement of tissue structure during inflammation in adipose tissue [151]. A whole-genome expression profile of liver, SM, SAT, and VAT was carried out in metabolically healthy obese and metabolically unhealthy individuals by using the weighted gene co-expression network analysis (WGCNA) to build within- and inter-tissue gene networks. In line with other studies, IL-1β was co-expressed with other genes of insulin-related pathways across tissues in metabolically unhealthy obese compared to metabolically healthy obese people, indicating a potential role in obesity [152].

Concerning the GO term *adherent junctions*, *TCF7L2* is one of the top *loci* associated with increased T2D risk [90]; epigenetic modifications showed by EWAS reporting lower methylation in SAT in T2D individuals [122], and association with BMI [123] and insulin resistance [124]. Therefore, it seems that *TCF7L2* in adipose tissue may have an impact on diabetes susceptibility and inflammation. Indeed, higher expression of *TCF7L2* was observed in adipose tissue of mice, associated with *NNAT* expression. Since SNPs in *NNAT* are found in obese humans, altered *NNAT* function in adipose tissue indicates an important function in adipocyte metabolism and inflammation [149]. In this study, they also identified that *LGALS12* and *IRF1* expression, a gene related to adipogenesis, co-varies with *NNAT* [149]. In relation to body weight, dietary intervention has an effect on weight loss and in gene expression patterns in adipose tissue. An adipose tissue transcriptome identified *EGFL6* and *FSTL3* differentially expressed in obese individuals who either maintained weight loss or regained weight during a dietary intervention [143]. 

*Focal adhesions* are large macromolecular assemblies through which mechanical force and regulatory signals are transmitted between the ECM and interacting cells. Thus, focal adhesion proteins are essential for embryogenic development and/or normal tissue and organ function, thereby elucidation of the mechanism represents a major challenge [155]. In adipose tissue, focal adhesions are implicated in the adipocyte function and insulin action. In particular, focal adhesion kinase (FAK), the central kinase regulating integrin signaling, regulates insulin sensitivity through adipocyte survival, establishing a link between the cell–ECM adhesion processes [156]. In this context, *JUN* expression, which co-localizes with *FAK* via a Ras/Rac1/Pak1/MAPK kinase 4 pathway, has been studied in the adipose tissue revealing an implication of *JUN* in the ECM remodeling in adipose tissue and metabolic function [157]. 

In the *endothelial cell migration* category, we gathered all genes regulating the orderly movement of endothelial cells into the ECM to form an endothelium. As a result, this category included *SIRT1* and *GATA3*. *SIRT1* expression was downregulated in the human subcutaneous adipocytes after irisin treatment, indicating a putative decrease in the adipocyte function [153]. In the study of Li et al. (2010), they identified *GATA3* is highly expressed in adipocytes, associated with *NNAT* expression, revealing its role in the inflammatory process in adipose tissue [149]. 

In the *ECM assembly and disassembly* category, we collected all genes regulating the assembly of ECM components and produced by cells, which are crucial for cellular differentiation, tissue morphogenesis, and physiological remodeling in adipose tissue. As a result, this category included six *loci* (*FURIN, MMP8, MMP13, FZD4, TLR3,* and *TLR4*). A recent study demonstrated that irisin modulates genes associated with severe coronavirus disease 19 (COVID-19) outcomes in human subcutaneous adipocytes. In particular, *FURIN* regulates angiotensin-converting enzyme 2 (ACE2), which is implicated in the viral infection, and irisin treatment decreases *FURIN* expression, establishing a link between ECM organization, irisin, and the putative response to viral infection in obesity conditions [153]. *FZD4* is a gene involved in adipogenesis and is differentially expressed in adipocytes, and its expression co-varies with *NNAT* expression [149]. *MMP8* and *MMP13* expressions are upregulated in adipose-stem cells (ASCs) from VAT compared to ASCs from SAT, suggesting that the fat depot-specific gene signatures of ASCs may contribute to the distinct patterns of ECM remodeling and tissue function in SAT and VAT [148]. Finally, *TLR4*, a member of toll-like receptor family involved in the inflammatory cytokine production, is increased in the presence of lipopolysaccharide or palmitic acid, and elevated glucose level conditions acts in concert to upregulate osteopontin expression by mononuclear cells through an *IL-6*-mediated mechanism [154]. 

ECM of SM is dysregulated in obesity and it is associated with insulin resistance, where collagens are upregulated in SM after 10% weight gain. Although the evidence of gene expression patterns in muscle is limiting compared to adipose tissue, it is well established a link between SM ECM remodeling and development of metabolic dysfunction [158]. A cohort study in healthy individuals showed an upregulation of *COL1A1*, *COL3A1,* and *MMP2* mRNA levels in SM 28 days after overfeeding, with no changes in *MMP9*. Microarray-based tests revealed alterations in pathways related to *ECM receptor interactions*, *focal adhesion* and *adherens junction*, elucidating new evidence linking SM ECM remodeling and obesity-related insulin resistance [159]. Insulin sensitivity and inflammation occur in SM in obesity by increasing immune cell infiltration and proinflammatory activation in intermyocellular and perimuscular adipose tissue. Moreover, macrophages participate in SM repair and regeneration by modulating inflammation, stem cells, cytokines, growth factors, and ECM. Indeed, endurance exercise training induces changes in M2 macrophages, which are positively associated with changes in ECM genes and decreased *IL6* expression after training [160]. 

Overall, microarrays and transcriptomic technologies have identified the role of several genes in adipose and SM ECM processes and structures and their implication in the different cell signaling pathways, which are disturbed in obesity and cause diverse complications in the metabolic homeostasis, and therefore, in the systemic metabolism. 

## 7. Conclusions and Further Perspectives

Omics-based technologies and biomarkers provide a great advance in our knowledge on the etiology of diseases, including obesity ant its metabolic complications; and this brings considerable potential in identifying effective public health strategies that pave the way towards patient stratification and precision prevention [161]. Since the 1970s, evidence on genetic origins of obesity uncovered that the pathogenesis of obesity is far more complex than just a dysregulation of energy balance, highlighting the importance of the gene-environment interactions by epigenetic modifications. In this context, novel etiological insights and targets are discovered through the omics approaches in order to understand the molecular mechanism of adipose and muscle ECM remodeling in obesity. Indeed, adipose tissue and muscle ECM remodeling plays a crucial role in the local and systemic metabolism, and understanding the crosstalk between both metabolic tissues is a challenge to develop new strategies against obesity. Here, we show, by exploring genomic (GWAS) epigenomic (EWAS) and transcriptomic (RNA-seq and cDNA microarrays) analysis, that *TCF7L2* is one of the top ECM-related *loci* associated with T2D risk and obesity. Interestingly, a reported sequence variation, DNA methylation (observed in three different studies) and differential gene expression of the *TCF7L2* region in SAT has been identified. Hence, *TCF7L2 loci* in adipose tissue may have an impact in diabetes susceptibility and inflammation. Furthermore, important genes implicated in adipocyte metabolism such as *ADIPOQ, CD36, PPARG,* and *IL6* exhibit genetic and epigenetic patterns associated with obesity in adipose tissue. We also found genetic variations and differential gene expression in adipose tissue for ECM genes such as *SIRT1*, a NAD-dependent deacetylase involved in cellular regulation. Finally, a hyper-methylation of *COL5A1* in VAT of women with insulin resistance was reported and a different gene expression pattern in VAT was also observed (Figure 1). Given the importance of muscle ECM regulating the insulin sensitivity and systemic metabolism, omics studies should address the molecular mechanism behind that in obesity. Nevertheless, according to the evidence encountered during the review, SM omics approaches are scarce. Despite the current evidence of genomic approaches in adipose tissue, still further studies are needed and larger prospective cohorts to validate findings and determine biomarkers, and even more in the case of SM. 

## Figures and Tables

**Figure 1 ijms-22-02756-f001:**
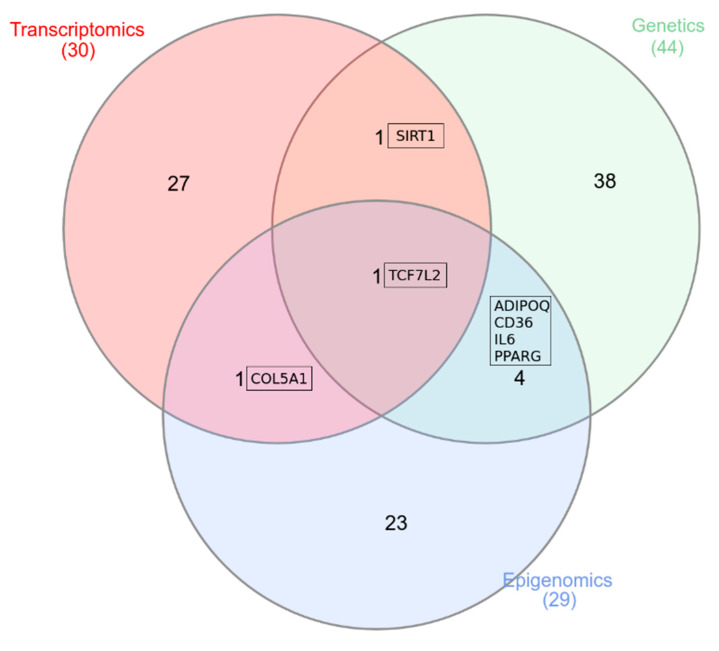
Venn diagram collecting the genes reported in omics approaches in adipose tissue. This figure shows the overlapping between genetics, epigenomics, and transcriptomic unveiling the role of ECM *loci* in obesity and metabolic comorbidities. *ADIPOQ*, adiponectin, C1Q, and collagen domain-containing; *CD36*, cluster of differentiation 36; *COL5A1*, collagen type V alpha 1 chain; *IL6*, interleukin-6; *PPARG*, peroxisome proliferator-activated receptor gamma; *SIRT1*, sirtuin-1; *TCF7L2*, transcription factor 7 like 2.

**Table 1 ijms-22-02756-t001:** Whole-genome genetics approaches reveal the implication of extracellular matrix (ECM) in obesity and metabolic dysfunction.

Cell Signaling/ Term	Gene Symbol and Name	Population	Study/Technology	Author
*Collagens*				
Collagen-containing extracellular matrix	*ADIPOQ* (Adiponectin)	1341 Framingham Heart Study participants in 310 families	GWAS on adiposity	Fox et al. 2007 [62]
Collagen-containing extracellular matrix	*ADIPOQ* (Adiponectin)	1156 Mexican mestizos with obesity and 473 normal weight controls	Candidate gene approach on obesity	León-Mimila et al. 2013 [63]
Collagen-containing extracellular matrix	*APOE* (Apolipoprotein E)	GIANT cohort (BMI as measure of obesity, *n* = 123,865)	Meta-analysis of GWAS on obesity	Hinney et al. 2014 [64]
Collagen-containing extracellular matrix	*APOE* (Apolipoprotein E)	132 European patients with severe HTG and 351 controls	GWAS on lipid traits	Wang et al. 2008 [65]
Collagen-containing extracellular matrix	*CBLN4* (Cerebellin-4)	1110 and morbid obesity bariatric patients from Taiwan	GWAS on obesity	Chiang et al. 2019 [66]
Collagen-containing extracellular matrix	*COL4A1* (Collagen alpha-1(IV) chain)	815 Hispanic children	Candidate gene approach on obesity	Comuzzie et al. 2012 [67]
Collagen-containing extracellular matrix	*COL6A5* (Collagen alpha-5(VI) chain)	5049 Europeans participants	GWAS on adiposity	Namjou et al. 2013 [68]
Collagen-containing extracellular matrix	*CTSS*	21,969 individuals from diverse ethnic populations	Meta-analysis of GWAS on obesity	Pei et al. 2014 [69]
	(Cathepsin S)			
Collagen-containing extracellular matrix	*F12*	242 monozygotic and 140 dizygotic twin pairs from a Northern Han Chinese population	GWAS on adiposity	Wu et al. 2018 [70]
	(Coagulation factor XII)			
Collagen-containing extracellular matrix	*F13A1* (Coagulation factor XIII A chain)	13 pairs of monozygotic Chinese twin pairs discordant for BMI and 77 controls	GWAS on adiposity	Naukkarinen et al. 2010 [71]
Collagen-containing extracellular matrix	*FRAS1* (Extracellular matrix protein FRAS1)	1060 obese cases and never-overweight controls	GWAS on uric acid levels	Li et al. 2013 [72]
Collagen-containing extracellular matrix	*ITIH4*	390 Mexican children with obesity and 405 normal weight controls	Candidate gene approach on obesity	Liu et al. 2019 [73]
	(Inter-alpha-trypsin inhibitor heavy chain H4)			
Dystroglycan complex	*SGCZ*	1335 African American subjects from GENOA cohort and 1224 from HyperGEN cohort	CNVs-GWAS on adiposity	Zhao et al. 2012 [74]
	(Zeta-sarcoglycan)			
Dystrophin associated glycoprotein complex	*SSPN*	27,350 African ancestry individuals	GWAS on adiposity	Liu et al. 2013 [75]
	(Sarcospan)			
*Cell adhesion*				
Cell adhesion molecule binding	*ADAM23* (Disintegrin and metalloproteinase domain-containing protein 23)	35,668 children from 20 studies in the discovery phase and 11,873 children from 13 studies in the replication phase	Meta-analysis of GWAS on adiposity	Felix et al. 2015 [76]
Cell matrix adhesion	*ADAMTS9* (A disintegrin and metalloproteinase with thrombospondin motifs 9)	5169 T2D cases and 4560 normal glycemic controls	GWAS on metabolic syndrome	Kong et al. 2015 [77]
Cell matrix adhesion	*ADAMTS9* (A disintegrin and metalloproteinase with thrombospondin motifs 9)	33,591 and 27,350 African ancestry individuals with waist circumference or waist-hip ratio	GWAS on adiposity	Liu et al. 2013 [75]
Cell matrix adhesion	*BCL6* (B-cell lymphoma 6 protein)	1028 unrelated European-American extremely obese females and normal weight controls	GWAS on lipid traits	Jiao et al. 2015 [78]
Regulation of cell adhesion	*C2CD4A* (C2 calcium-dependent domain-containing protein 4A)	10,701 non-diabetic adults of European ancestry under follow-up	GWAS on T2D	Strawbridge et al. 2011 [79]
Cell matrix adhesion	*CD36* (Platelet glycoprotein 4)	9973 European subjects	Meta-analysis of GWAS on obesity	Choquet et al. 2011 [80]
Cell adhesion molecule binding	*CDH12* (Cadherin-12)	1715 African American subjects	GWAS on adiposity	Ng et al. 2012 [81]
Cell adhesion molecule binding	*CDH18* (Cadherin-18)	1996 adult survivors of childhood cancer	GWAS on obesity	Wilson et al. 2015 [82]
Regulation of cell adhesion	*CELSR2* (Cadherin EGF LAG seven-pass G-type receptor 2)	40,473 African American, American Indian, Asian/Pacific Islander, European American, and Hispanic participants from 7 studies	Candidate gene approach on CRP levels	Kocarnik et al. 2014 [83]
Cell adhesion molecule binding	*COBLL1* (Cordon-bleu protein-like 1	1011 mediterranean subjects	GWAS on leptin levels	Ortega-Azorin et al. 2019 [84]
Regulation of cell adhesion	*CSK* (Tyrosine-protein kinase CSK)	1279 Japanese subjects (556 men and 723 women)	Candidate gene approach on adiposity	Hotta et al. 2012 [85]
Cell adhesion involved in heart morphogenesis	*FLRT2* (Leucine-rich repeat transmembrane protein FLRT2)	8089 African American women	GWAS and age of menarche	Demerath et al. 2013 [86]
Cell adhesion molecule binding	*IDH1* (Isocitrate dehydrogenase [NADP] cytoplasmic)	1263 Hispanic Americans	GWAS on adiposity	Gao et al. 2015 [87]
Cell adhesion molecule binding	*IGF1* (Insulin-like growth factor I)	5974 non-diabetic subjects	Meta-analysis of GWAS on IR	Hong et al. 2014 [88]
*Focal adhesion*				
Focal adhesion	*IGF1R* (Insulin like growth factor 1 receptor)	Obese patients after bariatric surgery	GWAS on adiposity	Rinella et al. 2013 [89]
Adherents junction	*INSR* (Insulin receptor)	23 obese PCOS women and 13 control women	GWAS on PCOS and obesity	Jones et al. 2015 [90]
Regulation of cell adhesion	*LEP* (Leptin)	1011 mediterranean subjects	GWAS on leptin levels	Ortega-Azorín et al. 2019 [84]
Regulation of cell adhesion	*LEP* (Leptin)	Swedish obese subjects after bariatric surgery	Candidate gene approach on adiposity	Sarzynski et al. 2011 [91]
Cell adhesion molecule binding	*NRXN3* (Neurexin-3)	520 cases (BMI > 35 kg/m(2)) and 540 control subjects (BMI < 25 kg/m(2)) of European ancestry	GWAS on obesity and related traits	Wang et al. 2011 [92]
Cell adhesion molecule binding	*OLFM4* (Olfactomedin-4)	35,668 children from 20 studies in the discovery phase and 11,873 children from 13 studies in the replication phase.	Meta-analysis of GWAS on adiposity	Felix et al. 2015 [76]
Cell adhesion molecule binding	*OLFM4* (Olfactomedin-4)	9377 children from the from the ALSPAC and the Raine Study	Meta-analysis of GWAS on adiposity	Warrington et al. 2015 [93]
Cell adhesion molecule binding	*PFKP* (ATP-dependent 6-phosphofructokinase, platelet type)	1000 US Caucasians	GWAS on obesity	Liu et al. 2008 [94]
Cell adhesion molecule binding	*PFKP* (ATP-dependent 6-phosphofructokinase, platelet type)	European Americans (*n* = 1496) and Hispanic Americans (*n* = 839)	GWAS on obesity	Scuteri et al. 2007 [95]
Regulation of cell adhesion	*RREB1* (Ras-responsive element-binding protein 1)	1060 obese cases and never-overweight controls	GWAS on uric acid levels	Li et al. 2013 [72]
Adherents junction	*TCF7L2* (transcription factor 7 like 2)	1235 Hispanic, 706 Asian, 1549 African American, and 2395 European American subjects from the Multi-ethnic Study of Atherosclerosis	GWAS on BMI	Salinas et al. 2016 [96]
Adherents junction	*TCF7L2* (transcription factor 7 like 2)	10,701 non-diabetic adults of European ancestry under follow-up	GWAS on T2D	Strawbridge et al. 2011 [79]
Positive regulation of endothelial cell migration	*MAP2K3* (Dual specificity mitogen-activated protein kinase kinase 3)	3562 American Indians	Candidate gene approach on adiposity	Bian et al. 2013 [97]
Positive regulation of endothelial cell migration	*MAP2K3* (Dual specificity mitogen-activated protein kinase kinase 3)	1975 Han Chinese type 2 diabetes patients	Candidate gene approach on Metabolic traits	Wei et al. 2015 [98]
*Endothelial Cell migration*				
Negative regulation of endothelial cell migration	*MAP2K5* (Dual specificity mitogen-activated protein kinase kinase 5)	Singaporean Chinese, Malay, and Asian-Indian populations (*n* = 10,482)	Meta-analysis of GWAS on adiposity	Dorajoo et al. 2012 [99]
Negative regulation of endothelial cell migration	*MAP2K5* (Dual specificity mitogen-activated protein kinase kinase 5)	8842 individuals from the Korean Association Resource data	Candidate gene approach on adiposity	Hong et al. 2012 [100]
Negative regulation of endothelial cell migration	*MAP2K5* (Dual specificity mitogen-activated protein kinase kinase 5)	Chinese children (*n* = 2977, 853 obese and 2124 controls	Candidate gene approach on adiposity	Lv et al. 2015 [101]
Negative regulation of endothelial cell migration	*MAP2K5* (Dual specificity mitogen-activated protein kinase kinase 5)	Childhood (3–17 years) and adulthood (18–45 years) follow-up for 658 subjects	Candidate gene approach on adiposity	Mei et al. 2012 [102]
Negative regulation of endothelial cell migration	*MAP2K5* (Dual specificity mitogen-activated protein kinase kinase 5)	2 030 unrelated Chinese children, including 607 normal weight, 718 overweight, and 705 obese	Candidate gene approach on adiposity	Wang et al. 2016 [103]
Negative regulation of endothelial cell migration	*MAP2K5* (Dual specificity mitogen-activated protein kinase kinase 5)	27,715; 37,691 and 17,642 individuals from three east Asian populations	Meta-analysis of GWAS on adiposity	Wen et al. 2012 [104]
Positive regulation of endothelial cell migration	*MET* (Hepatocyte growth factor receptor)	Search in GWAS catalog	GWAS on BMI and mental traits disorders	Hebebrand et al. 2018 [105]
Negative regulation of endothelial cell migration	*PPARG* (Peroxisome proliferator-activated receptor gamma)	927 non-diabetic African Americans	GWAS on IR	Chen et al. 2012 [106]
Negative regulation of endothelial cell migration	*PPARG* (Peroxisome proliferator-activated receptor gamma)	765 (556 males) from the NIMH CATIE sample	GWAS on drug-induced weight-regain	Corfitsen et al. 2020 [107]
Negative regulation of endothelial cell migration	*PPARG* (Peroxisome proliferator-activated receptor gamma)	1156 Mexican mestizos with obesity and 473 normal weight controls	Candidate gene approach on obesity	León Mimila et al. 2013 [63]
Negative regulation of endothelial cell migration	*PPARG* (Peroxisome proliferator-activated receptor gamma)	Review Swedish population	Candidate gene approach on obesity	Sarzynski et al. 2011 [91]
Positive regulation of endothelial cell migration	*PROX1* (Prospero homeobox protein 1)	964 chinese pregnant women with GMD and 1021 chinese pregnant women with normal glucose tolerance	GWAS on GMD	Cao et al. 2020 [108]
Positive regulation of endothelial cell migration	*PROX1* (Prospero homeobox protein 1)	756 individuals from a Mongolian sample	GWAS on obesity	Kim et al. 2013 [109]
Positive regulation of endothelial cell migration	*SIRT1* (Sirtuin 1)	3501 Pima Indians	Candidate gene approach on IR and T2D	Dong et al. 2011 [110]
*ECM assembly and disassembly*				
Extracellular matrix structural constituent	*GP2* (Pancreatic secretory granule membrane major glycoprotein GP2)	27,715; 37,691 and 17,642 individuals from three east Asian populations	Meta-analysis of GWAS on adiposity	Wen et al. 2012 [104]
Extracellular matrix disassembly	*IL6* (Interleukin-6)	16,088 postmenopausal women stratified by obesity status	GWAS on Proinflammatory Cytokines	Jung et al. 2020 [111]
Extracellular matrix	*LINGO2* (Leucine-rich repeat and immunoglobulin-like domain-containing nogo receptor-interacting protein 2)	100,418 adults from the single large multi-ethnic Genetic Epidemiology Research on Adult Health and Aging (GERA) cohort	Review of GWAS on adiposity	Speakman et al. 2013 [112]
Extracellular matrix disassembly	*SH3PXD2B* (SH3 and PX domain-containing protein 2B)	German case control population of 487 extremely obese children and adolescents and 442 healthy lean individuals; and an adult population of 1644 individuals from the German population-based study (KORA)	Candidate gene approach on obesity	Vogel et al. 2009 [113]
Extracellular matrix organization	*SLC2A10* (Solute carrier family 2, facilitated glucose transporter member 10)	56,000 unrelated individuals of several ethnics and cohorts	GWAS on adiposity	Hoggart et al. 2014 [114]

Abbreviations: BMI, body mass index; CNV, copy number variation; ECM, extracellular matrix; GMD, Gestational diabetes mellitus; GWAS, genome-wide association studies; HTG, hypertriglyceridemia; IR, insulin resistance; SNP, single-nucleotide polymorphism; T2D, type 2 diabetes.

**Table 2 ijms-22-02756-t002:** DNA methylation approaches reveal the implication of ECM in obesity and metabolic dysfunction.

Cell Signaling/Term	Gene Symbol and Name	Study/Tissue/Technology	Author, In Vitro/In Vivo
*Collagens*			
Collagen-containing extracellular matrix	*ADIPOQ* (Adiponectin)	Epigenome of human AT is affected by dietary fat composition and overfeeding in RCT/SAT/Array EWAS Illumina	Perfilyev et al. 2017, [115] In vivo
Collagen-containing extracellular matrix	*COL11A2* (Collagen alpha-2(XI) chain)	Severely obese men discordant for Metabolic syndrome/VAT/Infinium Human Methylation 450K Beadchip (Illumina)	Guénard et al. 2017, [116] In vivo
Collagen-containing extracellular matrix	*COL11A2* (Collagen alpha-2(XI) chain)	DNA methylation pattern in VAT differentiates insulin-resistant from insulin-sensitive obese subjects/Array EWAS Infinium Human Methylation 450K BeadChips (Illumina)	Crujeiras et al. 2016, [117] In vivo
Collagen-containing extracellular matrix	*COL23A1* (Collagen alpha-1(XXIII) chain)	An epigenetic signature in association with nicotinamide N-methyltransferase gene expression VAT/Infinium Human Methylation 450K BeadChip array (Illumina)	Crujeiras et al. 2018, [118] In vivo
Collagen-containing extracellular matrix	*LGALS3BP* (Galectin-3-binding protein)	Methylation signatures associated with obesity traits using leukocyte DNA samples from 2097 African American adults/SAT/EWAS Infinium Human Methylation 450K Beadchip (Illumina)	Demerath et al. 2015, [119] In vivo
Collagen-containing extracellular matrix	*PLG* (Plasminogen)	Age, BMI, and HbA1c levels on DNA methylation and mRNA expression patterns in human AT biomarkers in blood/AT/Array EWAS Infinium Human Methylation 450K BeadChips (Illumina)	Rönn et al. 2015, [120] In vivo
Collagen-containing extracellular matrix	*S100A4* (Protein S100-A4)	Evaluate the association between S100A4 and different obesity and insulin resistance parameters/vAT/Array AffymetrixHG-V133Plus 2.0 and RT-qPCR	Anguita-Ruiz et al. 2020, [121] In vivo
*Cell adhesion*			
Adherents junction	*TCF7L2* (transcription factor 7 like 2)	DNA methylation data in AT from monozygotic twin pairs discordant for T2D and independent case control cohorts/AT/Array EWAS Infinium Human Methylation 450K BeadChips (Illumina).	Nilsson et al. 2014, [122] In vivo
Adherents junction	*TCF7L2* (transcription factor 7 like 2)	Six months exercise intervention/AT/Array EWAS Infinium Human Methylation 450K BeadChips (Illumina).	Rönn et al. 2013, [123] In vivo
Adherents junction	*TCF7L2* (transcription factor 7 like 2)	Differentially DNA-methylated genomic regions in mouse and then replicated in human/AT/Array Comprehensive High-throughput Array-based Relative Methylation (CHARM) 7.5 million CpG sites	Multhaup et al. 2015, [124] In vitro e In vivo
Cell adhesion molecule binding	*SPP1* (Osteopontin)	DNA methylation data in AT from monozygotic twin pairs discordant for T2D and independent case control cohorts/AT/Array EWAS Infinium Human Methylation 450K BeadChips (Illumina)	Nilsson et al. 2014, [122] In vivo
Cell adhesion molecule binding	*SPP1* (Osteopontin)	Age, BMI, and HbA1c levels on DNA methylation and mRNA expression patterns in human AT biomarkers in blood/AT/Array EWAS Infinium Human Methylation 450K BeadChips (Illumina)	Rönn et al. 2015, [120] In vivo
Cell adhesion molecule binding	*BAG3* (BAG family molecular chaperone regulator 3)	Gene expression and DNA methylation respond to both short- and long-term weight loss/SAT/Infinium HumanMethylation 450K BeadChip	Bollepalli et al. 2018, [125] In vivo
Cell adhesion molecule binding	*COL5A1* (Collagen alpha-1(V) chain)	The epigenetic signature of systemic insulin resistance in obese women/SAT and VAT/Array EWAS Infinium Human Methylation 450K BeadChips (Illumina).	Arner et al. 2016, [126] In vivo
Cell adhesion molecule binding	*ADAM2* (Disintegrin and metalloproteinase domain-containing protein 2)	Genome-wide DNA methylation pattern differentiates insulin-resistant from insulin-sensitive obese subjects/VAT/Array EWAS Infinium Human Methylation 450K BeadChips (Illumina).	Crujeiras et al. 2016, [117]
Cell matrix adhesion	*CD36* (Platelet glycoprotein 4)	Case control study in 2 independent cohorts of obese/non-obese individuals/SAT and VAT/EZ DNA Methylation kit (Zymo)	Keller et al. 2017, [127] In vivo/In silico
Cell matrix adhesion	*CD36* (Platelet glycoprotein 4)	Case control study in obese subjects (Obese+Obese T2D) to evaluate if extracellular factors in obesity epigenetically reprogram adipogenesis potential and metabolic function of preadipocytes/VAT/Array EWAS RRBS	Andersen et al. 2019, [128] In vivo
Cell matrix adhesion	*SFRP1* (Secreted frizzled-related protein 1)	Role of a DNA methyltransferase (Dnmt3a) in obese AT from transgenic mice overexpressing Dnmt3a/AT/Array GWAS Affimetrics Mouse Genome 430 2.0	Kamei et al. 2010, [129] In vivo/In vitro
Cell matrix adhesion	*CDK5* (Cyclin-dependent-like kinase 5)	Case control study/SAT/Affymetrix Human Gene 1.0 ST arrays and DNA methylation using Illumina 450K BeadChip arrays.	Gillberg et al. 2016, [130] In vivo
Cell matrix adhesion	*CD44* (CD44 antigen)	Genome-wide DNA methylation pattern in insulin-resistant from insulin-sensitive obese subjects/VAT/Array EWAS Infinium Human Methylation 450K BeadChips (Illumina).	Crujeiras et al. 2016, [117] In vivo
Regulation of cell adhesion	*HLA-DPB1* (HLA class II histocompatibility antigen, DP beta 1 chain)	Age, BMI and HbA1c levels on DNA methylation and mRNA expression patterns in human AT biomarkers in blood/AT/Array EWAS Infinium Human Methylation 450K BeadChips (Illumina)	Rönn et al. 2015, [120] In vivo
Regulation of cell adhesion	*MAPK7* (Mitogen-activated protein kinase 7)	A RCT to study if the epigenome of human AT is affected differently by dietary fat composition/SAT/Array (Infinium Human Methylation 450K BeadChip, Illumina)	Perfilyev et al. 2017, [115] In vivo
Regulation of cell adhesion	*IGFBP2* (Insulin-like growth factor-binding protein 2)	Epigenetic changes of the *IGFBP2* gene associated with obesity by DNA methylation and mRNA expression in adipocytes from different depots/SAT and VAT/Array (Bisulfite pyrosequencing, PyroMark Q96)	Zhang et al. 2019, [131] In vivo
Regulation of cell adhesion	*CCL2* (C-C motif chemokine 2)	Adipocyte-expressed 1CC genes linked to WAT inflammation and IR from obese individuals/SAT, isolated adipocytes and *in vitro* adipocytes/Array (GeneChip^®^ Human Transcriptome Array 2.0 (Affymetrix) and EZ DNA Methylation-Gold Kit	Petrus et al. 2018, [132] In vivo/In vitro
Tight junction	*CLDN1* (claudin 1)	Genome-wide DNA promoter methylation along with mRNA profiles in non-obese vs. obese individuals/SAT and VAT/Array (Infinium HumanMethylation450K BeadChips (Illumina)	Keller et al. 2017, [127] In vivo
*Focal adhesion*			
Focal adhesion	*AKT2* (AKT serine/threonine kinase 2)	Case control study in subjects subdivided according BMI aimed to identify AT dysfunction involved in decreasing insulin action in adipocytes/SAT/Array EWAS (Illumina HumanMethylation27 BeadChip) + TWAS (Affymetrix Human 1.0 or 1.1 ST arrays) + pyrosequencing + RT-PCR	Rydén et al. 2019, [133] In vivo
*ECM assembly and disassembly*			
Extracellular matrix	*PHOSPH+B33O1* (Phosphoethanolamine/phosphocholine phosphatase)	DNA methylation *loci* in blood DNA (*ABCG1, PHOSPHO1, SOCS3, SREBF1*, and *TXNIP*) as predictors of future T2D/AT, blood, human pancreatic islets, liver, and skeletal muscle/Array EWAS Illumina Infinium Human Methylation 450K Beadchip	Dayeh et al. 2016, [134] In vivo
Extracellular matrix assembly	*NOTCH1* (Neurogenic locus notch homolog protein 1)	Smoking-associated DNA methylation and gene expression variation in AT biopsies from 542 healthy female twins/SAT, blood/Array EWAS Illumina Infinium Human Methylation 450K Beadchip	Tsai et al. 2018, [135] In vivo
Extracellular matrix structural constituent	*CD4* (T-cell surface glycoprotein CD4)	DNA methylation data to identify leukocyte cell types in obesity/SAT, blood/Array EWAS Illumina Infinium HumanMethylation450K Beadchip	Chu et al. 2019, [136] In vivo
Extracellular matrix structural constituent	*CD4* (T-cell surface glycoprotein CD4)	Methylation signatures associated with obesity traits using leukocyte DNA samples from 2097 African American adults/SAT/EWAS Infinium Human Methylation 450K Beadchip (Illumina)	Demerath et al. 2015, [119] In vivo
Extracellular matrix disassembly	*IL6* (Interleukin-6)	Epigenome of human AT affected differently by dietary fat composition and general overfeeding in a randomized trial/SAT/Array EWAS Illumina Infinium Human Methylation 450K Beadchip	Perfilyev et al. 2017, [115] In vivo
Extracellular matrix organization	*TNF* (Tumor necrosis factor)	Role of a DNA methyltransferase (Dnmt3a) in obese AT from transgenic mice overexpressing Dnmt3a/AT/Array GWAS Affimetrics Mouse Genome 430 2.0	Kamei et al. 2010, [129] In vivo/In vitro
Extracellular matrix organization	*TNF* (Tumor necrosis factor)	A RCT to study if the epigenome of human AT is affected differently by dietary fat composition/SAT/Array (Infinium Human Methylation 450K BeadChip, Illumina)	Perfilyev et al. 2017 [115] In vivo
Extracellular matrix organization	*TNF* (Tumor necrosis factor)	Case control study in obese subjects (Obese+Obese T2D) to evaluate if extracellular factors in obesity epigenetically reprogram adipogenesis potential and metabolic function of preadipocytes/VAT/Array EWAS RRBS	Andersen et al. 2019 [128] In vivo
Negative regulation of endothelial cell migration	*PPARG* (Peroxisome proliferator-activated receptor gamma)	A RCT to study if the epigenome of human AT is affected differently by dietary fat composition/SAT/Array (Infinium Human Methylation 450K BeadChip, Illumina)	Perfilyev et al. 2017, [115] In vivo
Negative regulation of endothelial cell migration	*PPARG* (Peroxisome proliferator-activated receptor gamma)	DNA methylation data in AT from monozygotic twin pairs discordant for T2D and independent case control cohorts/AT/Array EWAS Infinium Human Methylation 450K BeadChips (Illumina)	Nilsson et al. 2014, [122] In vivo
Negative regulation of endothelial cell migration	*PPARG* (Peroxisome proliferator-activated receptor gamma)	Genome-wide DNA promoter methylation along with mRNA profiles in non-obese vs. obese individuals/SAT and VAT/Array (Infinium HumanMethylation450K BeadChips (Illumina)	Keller et al. 2017, [127] In vivo/In silico
*Regulation of cytoskeleton*			
Regulation of actin cytoskeleton	*ARHGEF1* (Rho guanine nucleotide exchange factor 1)	Gene expression and methylation data from diabetic and healthy individuals/VAT/Array EZ-DNA methylation kit and Illumina Human Methylation 450K BeadChip	Wang et al. 2018, [65] In vivo
Regulation of actin cytoskeleton	*ARPC3* (actin-related protein 2/3 complex subunit 3)	Association of CpG-SNPs located within ARPC3, which is linked to adipogenesis/VAT/Array High-throughput array technology QuantStudio 12K Flex System	Toro-Martín et al. 2016, [137] In vivo

Abbreviations: AT, adipose tissue; ECM, extracellular matrix; EWAS, epigenome-wide association studies; RCT, randomized controlled trial; RRBS, Reduced Representation Bisulfite Sequencing; SAT, subcutaneous adipose tissue; SNP, single-nucleotide polymorphism; T2D, type 2 diabetes; VAT, visceral adipose tissue.

**Table 3 ijms-22-02756-t003:** Integrative analysis of gene expression patterns revealing ECM implication in obesity and metabolic dysfunction.

Cell Signaling/Term	Gene Symbol and Name	Study/Tissue/Technology	Author, In Vitro/In Vivo
*Collagens*			
Collagen-containing extracellular matrix	*ASPN* (Asporin)	Maintained weight loss or regained weight/AT/Microarray	Bolton et al. 2017, [143] In vivo
Collagen-containing extracellular matrix	*COL24A1* (Collagen alpha-1(XXIV) chain)	Gene × Environment analyses/AT/Microarray	Hong et al. 2017, [144] In vivo
Collagen-containing extracellular matrix	*COL6A1* (Collagen alpha-1(VI) chain)	ECM remodeling during the first year of bariatric surgery/SAT/Microarray and RT-PCR	Liu et al. 2016, [145] In vivo
Collagen-containing extracellular matrix	*COL6A2* (Collagen alpha-2(VI) chain)	ECM remodeling during the first year of bariatric surgery/SAT/Microarray and RT-PCR	Liu et al. 2016, [145] In vivo
Collagen-containing extracellular matrix	*EFEMP1* (EGF-containing fibulin-like extracellular matrix protein 1)	Copy number variants to early-onset obesity/SAT/Microarray	Petterson et al. 2017, [146] In vivo
Collagen-containing extracellular matrix	*ELN* (Elastin)	ECM remodeling during the first year of bariatric surgery/SAT/Microarray and RT-PCR	Liu et al. 2016, [145] In vivo
Collagen-containing extracellular matrix	*LOXL2* (Lysyl oxidase homolog 2)	*In vitro* 3D culture system/AT/Microarray and RT-PCR	Pellegrinelli et al. 2014, [147] In vitro
Collagen-containing extracellular matrix	*MMP8* (Neutrophil collagenase)	Gene signature and ECM remodeling/AT/RNA-seq and RT-PCR	Tokunaga et al. 2014, [148] In vitro
Collagen-containing extracellular matrix	*SOD3* (Superoxide dismutase)	Bio-informatics analysis/WAT of C57BL/6J mice/Microarray of four datasets	Li et al. 2010, [149] In vivo
Collagen-containing extracellular matrix	*SPARC* (Secreted Protein Acidic And Cysteine Rich)	Maintained weight loss or regained weight/AT/Microarray	Bolton et al. 2017, [143] In vivo
Collagen-containing extracellular matrix	*TGFB1* (Transforming growth factor beta-1 proprotein)	Maintained weight loss or regained weight/AT/Microarray	Bolton et al. 2017, [143] In vivo
*Cell adhesion*			
Adherents junction	*TCF7L2* (transcription factor 7 like 2)	Bio-informatics analysis/WAT of C57BL/6J mice/Microarray of four datasets	Li et al. 2010, [149] In vivo
Cell adhesion molecule binding	*ANXA1* (Annexin A1)	VAT from obese prepubertal children/Microarray	Aguilera et al. 2015, [150] In vivo
Cell adhesion molecule binding	*COL5A1* (Collagen alpha-1(V) chain)	SAT and VAT microarray	Arner et al. 2016, [126] In vivo
Cell adhesion molecule binding	*EGFL6* (Epidermal growth factor-like protein 6)	Maintained weight loss or regained weight/Adipose/Microarray	Bolton et al. 2017, [143] In vivo
Regulation of cell adhesion	*FSTL3* (Follistatin-related protein 3)	Dietary intervention/AT/Microarray and RT-PCR	Bolton et al. 2017, [143] In vivo
Cell adhesion molecule binding	*ICAM1* (Intercellular adhesion molecule 1)	Culture of human Adipocytes exposed to IL1B/AT/Microarray	Kępczyńska et al. 2017, [151] In vitro
Cell adhesion molecule binding	*IL1B* (Interleukin-1 beta)	Culture of human Adipocytes exposed to IL1B/AT/Microarray	Kępczyńska et al. 2017, [151] In vitro
Cell adhesion molecule binding	*IL1B* (Interleukin-1 beta)	Inter-Tissue Gene Co-Expression Networks liver/muscle, SAT, and VAT/HumanOmni-BeadChips (Illumina)	Kogelman et al. 2016, [152] In vivo
Regulation of cell adhesion	*IRF1* (Interferon regulatory factor 1)	Bio-informatics analysis/WAT of C57BL/6J mice/Microarray of four datasets	Li et al. 2010, [149] In vivo
Cell adhesion molecule binding	*LGALS12* (Galectin-12)	Bio-informatics analysis/WAT of C57BL/6J mice/Microarray of four datasets	Li et al. 2010, [149] In vivo
Cell adhesion molecule binding	*PTPRJ* (Receptor-type tyrosine-protein phosphatase eta)	SAT and VAT microarray	Arner et al. 2016, [126] In vivo
Cell adhesion molecule binding	*VCAM1* (Vascular cell adhesion protein 1)	Culture of human Adipocytes exposed to IL1B/AT/Microarray	Kępczyńska et al. 2017, [151] In vitro
*Focal adhesion*			
Focal adhesion	*JUN (Jun proto-oncogene, AP-1 transcription factor subunit)*	Bio-informatics analysis/WAT of C57BL/6J mice/Microarray of four datasets	Li et al. 2010, [149] In vivo
*ECM assembly and disassembly*			
Extracellular matrix disassembly	*FURIN*	Culture human subcutaneous adipocytes/AT/RNA-seq	de Oliveira et al. 2020, [153] In vitro
Extracellular matrix cell signaling	*FZD4* (Frizzled-4)	Bio-informatics analysis/WAT of C57BL/6J mice/Microarray of four datasets	Li et al. 2010, [149] In vivo
Extracellular matrix disassembly	*MMP13* (Matrix metalloproteinase-13)	ASCs isolated from subcutaneous and visceral adipose tissues/AT/RNA-seq and qRT-PCR	Tokunaga et al. 2014, [148] In vivo
Extracellular matrix	*TLR3* (Toll-like receptor 3)	Culture human subcutaneous adipocytes/AT/RNA-seq	de Oliveira et al. 2020, [153] In vivo
Matrix metallopeptidase secretion	*TLR4* (Toll-like receptor 4)	Culture of mononuclear cells with adipocytes/AT/RT-PCR and PCR array	Samuvel et al. 2010, [154] In vivo
*Endothelial Cell migration*			
Positive regulation of endothelial cell migration	*GATA3* (Trans-acting T-cell-specific transcription factor GATA-3)	Bio-informatics analysis/WAT of C57BL/6J mice/Microarray of four datasets	Li et al. 2010, [149] In vivo
Positive regulation of endothelial cell migration	*SIRT1* (NAD-dependent protein deacetylase sirtuin-1)	Culture human subcutaneous adipocytes/AT/RNA-seq	de Oliveira et al. 2020, [153] In vivo

Abbreviations: ASCs, adipose-derived stem cells, ECM, extracellular matrix, SAT, subcutaneous adipose tissue; SNP, single-nucleotide polymorphism; T2D, type 2 diabetes; VAT, visceral adipose tissue; WAT, white adipose tissue.

## Data Availability

Data sharing not applicable. No new data were created or analyzed in this study. Data sharing is not applicable to this article.

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
