# Peer review of "Omics Approaches in Adipose Tissue and Skeletal Muscle Addressing the Role of Extracellular Matrix in Obesity and Metabolic Dysfunction"

_ijms, 2021, doi:10.3390/ijms22052756_

Round 1
Reviewer 1 Report
The aim of the paper is to explain the molecular basis of ECM in white adipose tissue and skeletal muscle remodeling in obesity and the metabolic complications. This paper is written well. However, many reports are discussing about interstrain differences in mice in the severity of ECM remodeling during obesity development by a high fat diet. In other words, ECM remodeling is also dependent on the diet and how long the mice were given. The authors have to show at least the mouse strain and condition of diet (how much calories from fat).
Author Response
We would like to thank the reviewer for her/his overall positive assessment of our work. We now have included information regarding mouse strain and condition of diet (including % of fat) reported in the studies through the manuscript.
Reviewer 2 Report
This review paper by Anguita-Ruitz et al describes the literature reporting alterations in adipose tissue and skeletal muscle extracellular matrix (ECM) associated with obesity and metabolic dysfunction using omics approaches.
I feel that the writing is quite dense. As the molecules/genes have already been organised according to ECM gene ontology terms and their nature in the tables, it would make it easier for readers to appreciate the implications of alterations in these molecules/genes if the gene/molecules are compiled in sub-sections according to their implications in the text – eg., those associated with fat mass/BMI, those associated with dysregulation in glucose metabolism, with inflammation etc.
It should also be commented that the findings by themselves are associations and do not necessarily mean causative process, can reflect reactive processes, and that full mechanistic insights require co-ordinated sets of data at multiple time points collected from a number of relevant target tissues and require functional studies. Please discuss (also use * symbol to indicate in the table) which of the molecules/genes have been tested by the full characterisation studies to confirm their role in disease causation.
Please carefully review minor spelling and grammatical errors.
Author Response
This review paper by Anguita-Ruitz et al describes the literature reporting alterations in adipose tissue and skeletal muscle extracellular matrix (ECM) associated with obesity and metabolic dysfunction using omics approaches. I feel that the writing is quite dense. As the molecules/genes have already been organised according to ECM gene ontology terms and their nature in the tables, it would make it easier for readers to appreciate the implications of alterations in these molecules/genes if the gene/molecules are compiled in sub-sections according to their implications in the text – eg., those associated with fat mass/BMI, those associated with dysregulation in glucose metabolism, with inflammation etc.
We would like to thank to the reviewer for his/her assessment. We agree with the reviewer that writing is a bit dense and we organized the tables according to ECM gene ontology. However, subsections in the text would make it even longer, since we described the text according to the different GO terms (collagens, cell adhesion…) in order to make it easier to follow the information collected in the tables.
It should also be commented that the findings by themselves are associations and do not necessarily mean causative process, can reflect reactive processes, and that full mechanistic insights require co-ordinated sets of data at multiple time points collected from a number of relevant target tissues and require functional studies. Please discuss (also use * symbol to indicate in the table) which of the molecules/genes have been tested by the full characterisation studies to confirm their role in disease causation.
We agree with the reviewer that associations do not mean causative process. Here, we have described the associations from the different studies using omics approaches, and no causal relationship has been discussed. In order to avoid confusion, we would appreciate what specific sentence or paragraph should be changed. We have included the follow sentence in the text, page 5 “In the present section, we gathered all literature findings from GWAS experiments in which sequence variation for ECM-related loci has been linked to obesity or meta-bolic dysfunction, although associations do not mean causative process.” In addition, we have included in page 12 “In this section, we assembled all literature findings from EWAS experiments re-vealing the implication of ECM in obesity and metabolic dysfunction. We collected and organized according to ECM gene ontology terms, specifically focusing on biological process terms, but associations do not mean causative process (Table 2).”
Please carefully review minor spelling and grammatical errors.
Thanks for the revision. We have checked carefully the whole manuscript in order to avoid spelling and grammatical errors.
Reviewer 3 Report
There are many literature/web-resources/databases on adipose tissue and skeletal muscle ECM focused on omics approaches (for example, MatrisomeDB). This article focuses primarily on GWAS and EWAS observations on ECM-related alterations in WAT and SM linked with metabolic dysfunction and obesity.
There are a few issues and suggestions that need to be resolved in order to improve the quality of the paper
- molecular mechanism of ECM remodeling in WAT and SM needs to explain more.
- the methodology section should be concise.
- In the study selection heading, the phrase "All data was obtained by...." is already stated in the author's contribution heading, which I do not think is relevant here.
- heading 4 and legend of table 1 is same "Whole-genome genetics approaches reveal the implication of ECM in obesity and metabolic dysfunction".
- Table 1, six pages long, could be reorganized. Get it more compact and insightful.
- In the text, Figure 1 is not mentioned.
- Several ECM genes are common for adipogenesis and myogenesis, which should be included in the manuscript.
Author Response
There are many literature/web-resources/databases on adipose tissue and skeletal muscle ECM focused on omics approaches (for example, MatrisomeDB). This article focuses primarily on GWAS and EWAS observations on ECM-related alterations in WAT and SM linked with metabolic dysfunction and obesity.
There are a few issues and suggestions that need to be resolved in order to improve the quality of the paper
- molecular mechanism of ECM remodeling in WAT and SM needs to explain more.
We appreciate the reviewer comment. However, the aim of the present work was to include the omics approach addressing genetic, epigenetic, or gene expression mechanisms in WAT and SM in order to understand the molecular mechanism of ECM in metabolic dysfunction associated with the obesity development. The heading 2, which describe the basis structure and function of ECM is indeed a brief introduction to ECM remodeling without all the described details. The molecular mechanism of ECM remodeling has been reviewed elsewhere (Ruiz-Ojeda 2019, PMID: 31581657; Ahmad 2020, PMID: 32481704). We have included this information in the page 5: “The specific molecular mechanism of ECM remodeling has been reviewed elsewhere 60, 61”.
- the methodology section should be concise.
We agree. Nevertheless, this systematic review was performed according to the guidelines described by the Preferred Reporting Items for Systematic Review and Meta-Analysis (PRISMA), and all the bibliographic search, inclusion and exclusion criteria have been explained properly. We have reduced a bit the last part of the methods section.
- In the study selection heading, the phrase "All data was obtained by...." is already stated in the author's contribution heading, which I do not think is relevant here.
We agree that information through the manuscript should not be duplicate. Now, we have deleted this part in the methods section.
- heading 4 and legend of table 1 is same "Whole-genome genetics approaches reveal the implication of ECM in obesity and metabolic dysfunction".
This is the title of the paragraph 4 and, of course, the title is the same in the table since here is collected all the important information and references described in the section 4, which it is the easiest way to find some data quickly.
- Table 1, six pages long, could be reorganized. Get it more compact and insightful.
We agree that table 1 is too long. Collecting all information regarding this part (whole-genome genetics approaches revealing the implication of ECM in obesity and metabolic dysfunction) drives many references, which have been explained in the text accordingly. Now, we have summarized the table content in order to make it more compact and insightful.
- In the text, Figure 1 is not mentioned.
We apologized for this mistake. In principle, the figure 1 was mentioned in the text, but some typo happened. Now we have corrected properly.
- Several ECM genes are common for adipogenesis and myogenesis, which should be included in the manuscript.
We agree. We have included information regarding the ECM genes in relation to adipogenesis and myogenesis in the manuscript (page 5).
Round 2
Reviewer 3 Report
The authors replied well and incorporated almost all the suggestions in the revised manuscript.